# "A Little Flip Goes a Long Way"—The Impact of a Flipped Classroom Design on Student Performance and Engagement in a First-Year Undergraduate Economics Classroom

## Nadia Singh

Newcastle Business School, Northumbria University, Newcastle upon Tyne NE1 8ST, UK;
nadia.singh@northumbria.ac.uk

**Abstract:** The flipped classroom is gaining prominence as an active learning pedagogy to engage a new generation of students. However, all courses do not lend themselves to a fully flipped design and instructors are often reluctant to flip lectures due to the additional time and effort involved, especially so in case of technical subjects such as economics. This study experiments with a flipped classroom design in a first-year undergraduate economics course. The research was motivated by the fact that many undergraduate economics students do not engage with traditional lectures. They fail to acquire critical thinking, data handling and reasoning skills, which are thought to be at the core of the economics curriculum. In this flipped classroom format, traditional lectures were substituted with micro-lectures and the remaining class time was devoted to active learning pedagogies including quizzes, group work and student presentations. The full lectures were panopto recorded and put up on the e-learning site, Blackboard. In order to evaluate the effectiveness of the flipped classroom format, I compared the final exam scores of students in the flipped classroom with those in the control group, which followed a traditional lecture-based approach. The key results from the analysis revealed that students in the flipped classroom performed better in the final exams vis-à-vis students in the traditional classroom format. Furthermore, students in the flipped classroom format were 1.61 times less likely to fail in the module as compared to students in a traditional classroom format. This format also resulted in better student engagement, more flexibility and enhanced student–tutor interactions within the classroom.

**Keywords:** flipped classroom; engagement; flexibility; active learning pedagogies; micro-lecture; economics

## 1. Introduction

A major challenge in higher education teaching today is the development of appropriate learning pedagogies to engage Generation Z students. Generation Z represents a generation of students born between 1995 and 2012 [1]. They have been christened as "digital natives" by Prensky [2] to denote that this generation of students is more technologically sophisticated as compared to the previous generations and have a natural aptitude for new devices and online interactions. Some studies have shown that Generation Z students suffer from low attention spans and find it hard to engage with classroom lectures [3–5]. It is argued that due to their early experience with technology, these students have a preference for high-speed, discovery-based learning styles, which enables them to explore and test new ideas [6]. However, other scholars have argued that familiarity with technology is not universal, and is conditioned by socio-economic status, culture and early access to technology [7]. It has also been claimed that students' preference for technology-enhanced learning may be conditioned by their cognitive abilities and individual learning styles [6]. Thus, there is a need to go beyond simple

dichotomies evident in the "digital natives" debate and consider the complexities associated with higher education students' relationship with technology.

The traditional "chalk and talk" style of teaching is being increasingly regarded as "transmissive and passive, with little room for student participation, low student engagement and a learning environment supporting only a surface approach to learning" [8]. These deficiencies of "chalk and talk"-style approaches have not been sufficiently recognized in undergraduate teaching across many social science disciplines, including economics [9]. It has been estimated that undergraduate lecturers on an average spend 65–80% of class time on traditional lectures [10]. However, the dominant role of lecturing in higher education is being increasingly challenged. Studies have revealed that a significant proportion of undergraduate students only acquire conceptual knowledge of the subject in a traditional classroom format. They are not able to adequately develop critical thinking, data analysis and reasoning skills, which are thought to be at the core of the curriculum [11]. Employers also reveal that many business school graduates are not good at applying economic and finance principles to real-world issues [5].

It is thus crucial to experiment with student-centered, active learning pedagogies in higher education. One such active learning pedagogy is the "flipped" classroom model. A "flipped" classroom involves the replacement of traditional lectures with out of class delivery of course content through online resources. The in-class time is devoted to collaborative, hands-on problem solving and discussion [12]. The overarching aim of this form of learning is to create a student-centered learning environment and transform the "acquisition of foundational knowledge as an out of class activity" [13]. In contrast to traditional lectures, this form of learning requires students to be active participants in sharing, understanding and constructing new forms of knowledge [14]. In a "flipped" model, the role of the tutor is relegated to a "guide on the side" instead of a "sage on a stage" [15]. Thus, the tutor becomes more of a facilitator, rather than a content provider, which is the case in a traditional classroom lecture. As the focus of the learning process is shifted from the educator to the learner, students are able to take ownership of their learning process.

However, existing studies show that the impact of the flipped classroom on students' academic performance is rather mixed. Calimeris and Sauer [16] implemented a "flipped" classroom in a principles of micro-economics classroom and found that students in a "flipped" classroom scored significantly higher on the mid-term as well as final exams, compared to students in a non-flipped classroom. Some studies have reported that students in a "flipped" classroom model perform only moderately better than a traditional lecture-based format [17–21]. Other studies have shown contradictory results and revealed that the students in the flipped classroom performed poorly, as compared to a traditional lecture-based format [11,22].

With respect to evaluation of student engagement in a "flipped" classroom format, there is also considerable variation in results. Vasquez and Chiang [23] examined student satisfaction in a first-year undergraduate micro-economics classroom, comprising 900 students. They found that a majority of students reported satisfaction with the "flipped" learning design. These students reported that the "flipped" classroom format supported their learning needs better than a traditional lecture and regarded it as more interactive. Ageline and Garcia-Cabonell [24] demonstrated that a flipped classroom format implemented in an undergraduate English literature classroom helped to improve students' writing skills in terms of organization and linking of ideas. Other recent studies have revealed that in a flipped classroom format, students have higher levels of motivation, as well as autonomy and self-regulation [25,26]. Lage et al. [27] found that the "flipped" classroom helped to create more inclusivity in the classroom and support students with different learning styles. In another study, McLean and Attardi [28] analyzed student perceptions of a flipped classroom experience. Their findings revealed that students valued the role of the tutor as a "facilitator" rather than an "information deliverer". Students also valued peer learning and more classroom interaction greater, which resulted in a flipped classroom format. Similar results were reported by other scholars as well [29–32].

However, challenges remained with respect to students' engagement with the pre-class work and the assigned readings in a flipped classroom format. This was qualified by Heijstra and Siugroardottir [33] who implemented a "flipped" classroom format in a graduate-level research methods course. Their findings revealed that the student performance in a "flipped" classroom format depended crucially on their class preparation and the time spent viewing the recordings and the online material. Becker and Proud [19] also reported that the "flipped" classroom design was associated with higher student satisfaction only among students who had engaged with the pre-class work and were able to pick up the first-order cognitive skills comprising understanding and comprehension, due to reading the materials at home. Students who failed to engage or understand the pre-class videos became further disengaged in a "flipped" classroom format. Furthermore, Foster and Stagl [34] revealed that students considered the workload in a flipped classroom format as too high and burdensome. They felt that they could achieve the same learning outcomes in a traditional classroom with much less pressure.

The "flipped" classroom pedagogy is still in the nascent stage of development. Questions remain on the best practice models to be implemented in a "flipped" classroom format, which would result in better student outcomes as well as higher student engagement in the classroom. This factor has been highlighted by recent scoping studies on the "flipped" classroom design [35,36]. Some scholars also argue that many times tutors are reluctant to implement the flipped classroom model due to the additional time and costs involved [37]. Furthermore, the flipped classroom format does not give tutors adequate control over the classroom, and the synergies and tangents which emerge in a classroom lecture are often missing in a flipped classroom format [5]. Song et al. [38] stated in this context that "there is still a pressing new for studying HOW to implement the 'flipped' classroom and moreover to be able to connect this pedagogical design with evidence of advantages related to various aspects of student learning".

This paper attempts to contribute to this critical area and add to the existing literature on the subject. The research is motivated by the fact that there is still considerable ambiguity on the best practice model to be followed in case of a flipped classroom design. This inquiry will serve as an exemplar for other tutors trying to implement a "flipped" classroom model, especially so in the field of economics. The results from this study will also provide further evidence on the opportunities as well as contestations surrounding the implementation of the "flipped" classroom model.

The main research questions, which the study seeks to address, are:

(1) What is the best practice model to be followed in a flipped classroom model?
(2) Do students in a flipped classroom format perform academically better as compared to students in a traditional lecture-based format?
(3) What is the perception of students towards a flipped classroom format?

In this study, I analyze how a traditional principle of the economics classroom was converted into a flipped classroom format. In this format, the full lecture was offloaded on the online learning portal, Blackboard, in the form of panopto videos. The class time was devoted to a micro-lecture, followed by classroom activities such as quizzes, classroom discussion and student presentations. This classroom design enabled me to combine the pedagogical benefits of a traditional lecture and a flipped classroom. One could reinforce the key theoretical arguments in the micro-lecture, but at the same time reap the benefits of increased student participation from the flipped classroom format. I present in detail the design of the flipped classroom model and the process, which was implemented to evaluate student performance and perceptions of this model in the subsequent sections.

The remainder of the paper is organized as follows. Section 2 puts forth the theoretical framework that underpins the "flipped" classroom model. Section 3 outlines the flipped classroom design that was followed in this experiment. Section 4 compares the student performance in the flipped classroom with the control group where a traditional lecture-based approach was followed. Section 5 examines the student perception of the flipped classroom format through the module evaluation survey. Section 6 concludes.

## 2. Theoretical Framework

In this inquiry I analyzed the "flipped" classroom model in terms of two key learning perspectives—the social constructivist theory and the cognitive load theory. These are umbrella theories that put the students' interests, learning styles and abilities at the center of the learning process [39]. The constructivist theory of knowledge says that knowledge is a state of understanding which results from continuous interaction between the environment and the individual [40]. Students do not come to the classroom as blank slates, but as learners with their own prior experiences and perspective on the topic. When they encounter new forms of knowledge and new information on the given topic, it interacts with their prior knowledge of the subject and helps them to develop a unique understanding of the subject [41]. As such, knowledge is constantly constructed and re-constructed by individuals, on the basis of their existing knowledge as well as lived experiences of the individual, and interaction with peers [42].

A "flipped" classroom is a pedagogical space in which "direct instruction moves from an individual space to a group space, resulting in the group space becoming a dynamic interactive space" [43]. The "flipped" classroom involves a reversal of Bloom's taxonomy [44]. Students are able to do lower-level cognitive work (understanding and comprehension) outside class, while class time is devoted to higher-order cognitive tasks comprising problem solving, application of key concepts and analysis through collaborative group work, problem solving, quizzes and classroom discussion. The "flipped" classroom thus aligns well with the social constructivist approach and is a useful theoretical framework to assess student engagement and performance within the classroom model. It would be instructive to see the channels through which autonomy and peer learning influence student outcomes in this classroom design.

The other mechanism through which the "flipped" classroom impacts learning outcomes is through cognitive load. The cognitive load depends on: (1) learner outcomes; (2) learner's prior knowledge; (3) learner settings [45]. The cognitive load theory states that the working memory has a certain capacity and it experiences a number of "loads" during the learning process [35]. These comprise an intrinsic load (the core of the concept), extraneous load (the additional load that does not translate into learning) and germane load (the additional load that helps learning by leading to the production of schema) [46]. In a "flipped" classroom setting, the students are exposed to the materials before the class and they are therefore able to pace their learning and choose the learning strategy that is most conducive to their learning preference. As a result, they may be able to identify the intrinsic load in advance; at the same time, they may be able to reduce the extraneous component of their cognitive load [35].

These two theoretical frameworks informed the design, implementation and evaluation of the "flipped" classroom model. Within this conceptual framework, I linked the "flipped" classroom design to the key learning outcomes in the principles of economics course, which comprised (a) developing core competencies in economic theories and principles; (b) the ability to apply economic theories to real-world policy issues; (c) developing quantitative reasoning and problem-solving skills. The conceptual framework is presented in Figure 1 above. This framework illustrates how the "flipped" classroom room was designed, rooted in the key tenets of the cognitive load theory and the social constructivism paradigm to achieve the key learning outcomes in terms of understanding economic theories, applying theoretical principles and developing quantitative reasoning skills. With a view to achieve these learning outcomes, the "flipped" model had four key components: self-paced online learning, quizzes, pair and share activities, group discussions and presentations. These key activities are explained in detail in the following section.

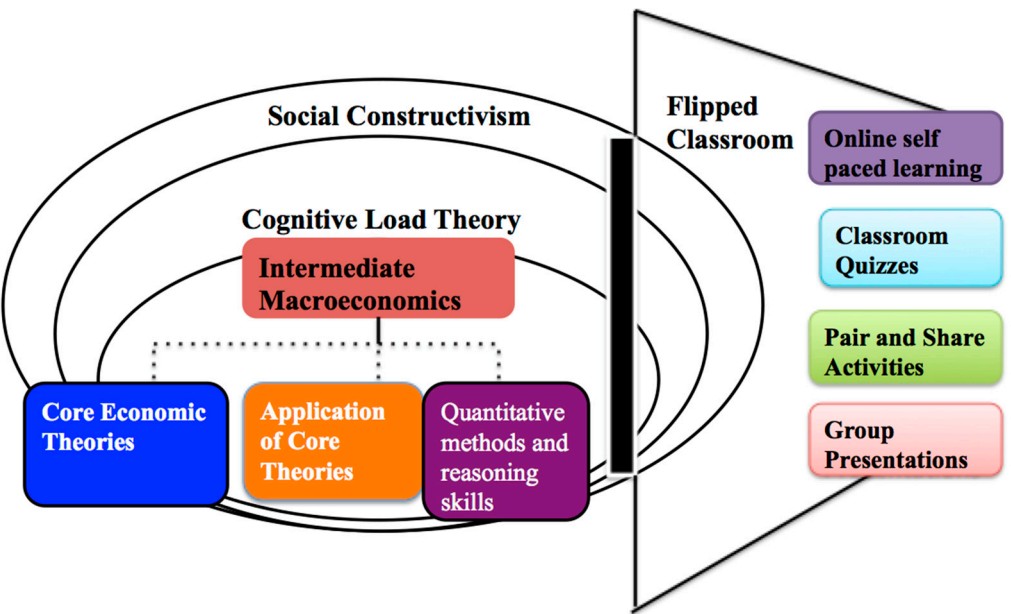

**Figure 1.** The conceptual framework of the design and implementation of the "flipped" classroom model in a principles of economics classroom (source: own compilation based on review of literature).

## 3. The "Flipped" Classroom Design

The participants in this course comprised 250 students enrolled in two sections of a principles of economics module. This was a compulsory course for all first-year students. The course was traditionally taught through a weekly two-hour lecture followed by small group seminars of one hour each. The lectures covered the theoretical aspects of the module, while the seminars provided skill training in these areas. Previous course evaluations revealed that students found the lecture content too heavy and difficult to engage with. They also expressed that they wanted to receive more hands-on support in the quantitative aspect of the module. The course redesign was thus motivated by the desire to fully engage students in the learning process and stimulate higher-order thinking and problem-solving skills among students through the use of creative technologies.

At the beginning of the semester, students were informed of the purpose of the study and were asked to provide informed consent. The sections were randomly assigned to be either the flipped classroom section ($n$ = 120) or the "traditional" section ($n$ = 130). Both the sections were taught by the same tutor. The traditional section continued to be taught through a weekly two-hour lecture followed by small group seminars of one hour each. The same seminars were conducted for the flipped classroom section as well. However, the lectures in the flipped classroom were replaced by an alternative format.

As shown in Figure 2, in the new format, all the in-class lectures were converted to self-paced online videos. I pre-recorded 24 lectures in the form of videos using Panopto software and uploaded them on the online learning platform, Blackboard. These lectures emphasized the critical concepts and theories, which students needed to learn prior to coming to class. These videos were interspersed with interactive learning exercises, which enabled students to regulate their own learning. The students could access these videos at any time on their computers or any internet-enabled device. They had the ability to pause, rewind and fast forward the videos and watch them at their own pace. Students were asked to view two videos each week with an average duration of 18.4 min (range of 15 to 29 min). Along with the pre-recorded videos, students were assigned background readings and textbook chapters each week. This constituted preparatory work for students prior to coming to class.

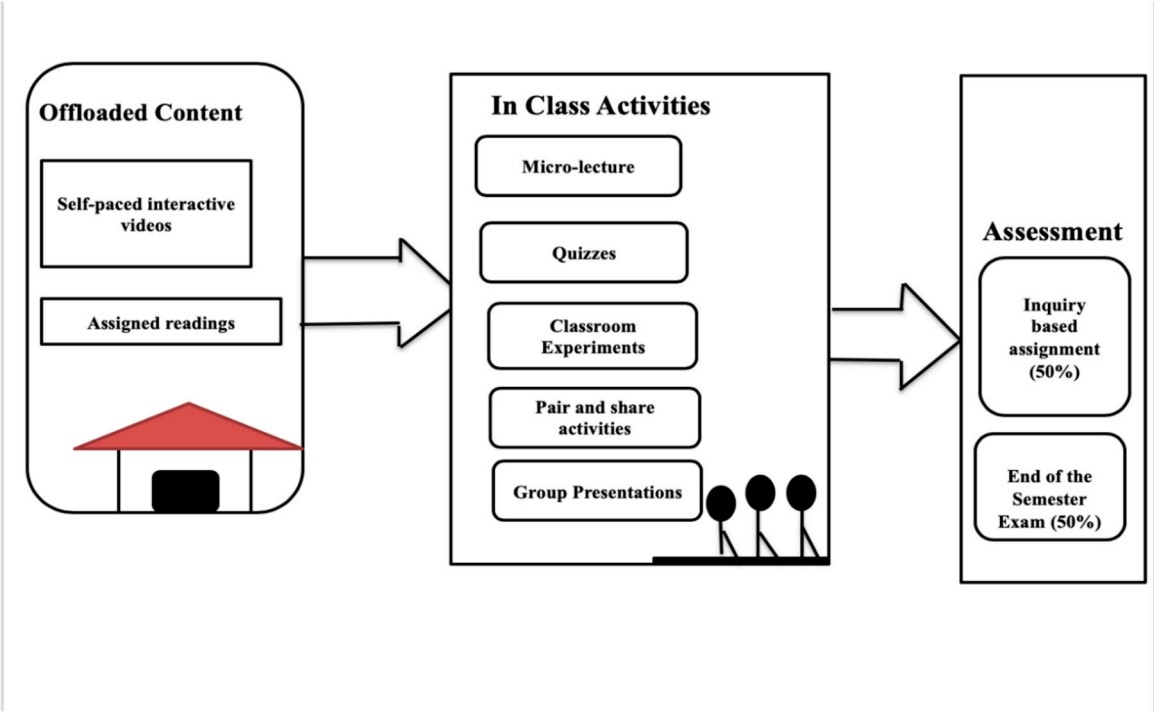

**Figure 2.** The flipped classroom design (own compilation).

The 2-h lecture was broken down into a number of activities in the flipped classroom format. The class began with a micro-lecture. These "micro-lectures" enabled me to direct and reinforce on the students' understanding of the online materials. I summed up the key aspects of the material, focusing on the graphs and derivation of equations. From my previous experience, I knew that students often struggle with this aspect of economic theory. This was followed by a Multiple Choice Question (MCQ) quiz using PollEverywhere software. This quiz comprised 10–15 multiple-choice questions. The students answered these questions through their hand-held devices and were allocated approximately 30 s per question. The responses were then analyzed and feedback was provided to students. Students were then put in pairs and asked to solve 2–3 numerical questions or a case study related to the pre-class classwork. (Pair and share activities represent a collaborative learning strategy where students work together to solve a numerical problem or answer an open ended question. This learning strategy encourages students to reflect critically on an issue and share their findings with others. It helps students to extend their conceptual understanding of an issue through clear examples, and gain practice in using other people's opinions to develop their own thoughts and ideas [47].)"

While the students were solving the questions, I went around the class and provided them with step-by-step guidance. Finally, a few pairs of students were invited to come forth and present the answers to class. I followed this up with my own feedback and guidance on the numerical questions. In some of the sessions the "pair and share activities were replaced by classroom games such as the market experiment for supply and demand. In the last part of the class, four groups of 4–5 students each were asked to present a summary and interpretation of the pre-assigned readings and answer students' questions on the topic. Each group was required to give presentations twice during the course of the semester.

Appropriate forms of assessment are considered crucial to the success of a "flipped" learning design [48]. In this flipped classroom design, I adopted two forms of assessment, which enabled me to evaluate students' understanding of the materials, their numeracy skills and their ability to apply economic theories and policies to real-world policy issues. These comprised an inquiry-based assignment (50%) and an end of semester exam (50%). The assignment required students to conduct independent research and apply economic theories covered in class to a real-world-problems.

The assignment tested the students' ability to understand and apply core economic theories. In the flipped classroom, pair and share activities and group presentations based on pre-assigned readings also contributed to achieving these key learning objectives and further students' understanding and application of key economic theories. The end of semester exam comprised ten MCQ-based questions and two essay type questions. The exam tested students' understanding and application of key economic theories, as well as their quantitative thinking, reasoning and problem-solving abilities. The classroom activities, such as quizzes, experiments, and solving numerical/case study questions collaboratively, helped to achieve these key learning outcomes during the course of the semester.

## 4. Outcomes of the Flipped Classroom Design

The outcomes of the "flipped" classroom model were examined on the basis of the student performance in this course, as well as student perceptions and evaluation of this model.

### 4.1. Data Sources

I compared students' grades in the two assessment tasks—i.e., the end of the semester exam and the inquiry-based assignment in the flipped classroom model with those in the treatment group where a traditional lecture-based approach was followed. In addition to this, I invited students in both the sections to participate in this research and fill in a short Qualtrics survey comprising ten questions to collect information on their ability to conduct independent study, number of absences from class, average time spent studying at home, whether they worked part-time during the semester and if they had studied economics at A level. The students were asked to fill this survey in one of the seminar sessions and were awarded an additional 2 marks in the final exam in order to incentivize them to complete the survey. Each student was provided with a participant sheet, detailing the purpose of the research, and were asked to sign an informed consent sheet. Students were also provided with the opportunity to ask questions about the research process and the purpose for which their individual data would be used. They were also informed that they may withdraw from participating in this project at any time if they so wish.

This form stated the purpose of the research projects and sought permission from the respondents to use their demographic data for the purpose of the study. Steps were taken to ensure that all students participating in the study were anonymized. A total of 104 out of 120 students in the flipped classroom section and 113 out of 130 students in the traditional classroom filled in the survey.

The student perception of the flipped classroom model was based on the teaching evaluation survey. This survey was conducted at the end of the semester and comprised 10 Likert-type questions and one open-ended question. Students in the flipped classroom section were invited to participate in this research and were given detailed information about the purpose of the research. Each student was provided with a participant information sheet and asked to sign an informed consent form which gave permission to the researcher to use their data and quotes in her study. Students rated statements about the "flipped" classroom on a scale ranging from 1 to 4. A value of 1 refers to "strongly disagree" and 4 refers to "strongly agree". In total, 82 out of the 120 students in the flipped classroom section completed the survey, which corresponded to 68.3% of the class. The students were asked to fill in the survey in class during one of the revision sessions.

In order to code the open-ended question, I read through the transcripts several times in order to immerse myself in the data. Open coding was used to determine what was represented in the text, and the phenomena was given a name [49]. Codes were grouped into categories and sub-categories to develop the initial coding frame and set rules. After the initial coding frame was developed, I provided clear descriptions of each category and set rules determining when and what data should be included under a category. The codes were then examined and re-organized and all transcripts were recoded using the final codes to establish consistency. To ensure the trustworthiness of the findings, I used peer examination [50] and asked another academic to comment on the plausibility of the results.

### 4.2. Modelling Student Performance in the Flipped Classroom

I employed a multiple regression analysis in order to evaluate the effects of a flipped design on students' academic performance. This technique is considered superior to a simple difference in means test between a flipped and non-flipped model as it allows to control for other demographic, parental and personal attributes that may affect a student's grades [51]. The regression model was specified as:

$$GRADE_i = a + bFLIP_i + DiY + U_i \tag{1}$$

(1)    $GRADE_i$ is the student's aggregate score in the final exam and the assignment.

(2)    i = 1.n. FLIP is a dummy variable with a value equal to 1 if the student was enrolled in the flipped classroom and 0 if the student is enrolled in the flipped classroom.

(3)    $D_i$ is a vector of academic controls including number of absences from class, average time spent devoted to independent studying and whether the student studied A level economics and if the student worked part-time during the semester.

In addition to this, I also analyzed whether a flipped classroom model reduced the probability of a student failing the course. In the UK higher education system, a student who gets an overall grade below 40% fails the course. Following Lombardini et al. [11], I employed a binary logit regression model. In this model, the dependent variable was coded as 1 if the student's overall grade was below 40% and 0 otherwise. The results of this binary regression model were used to predict the probability of the student failing the exam given specific demographic characteristics (gender, age, ethnicity and parental education) and academic characteristics (number of absences from class, average time spend on independent study, whether the student studied A level economics and if the student works part-time during the semester).

### 4.3. Results

Table 1 presents the summary statistics for the two sections—the flipped classroom (*n* = 104) and the traditional lecture-based classroom (*n* = 113). The differences between the two sections are determined to be significant on the basis of a t-test for difference in means. There are no significant differences between the two sections in the final grades for the assignment, while the students in the flipped classroom scored significantly higher in the end of the semester exam. The students in the two sections are similar in their academic characteristics. There are no significant differences between the two sections in terms of number of absences from class, the average time devoted to independent studying, taking economics at A level and working part-time during the semester. This shows that the two sections are relatively homogenous in terms of their observable characteristics.

The results of the Ordinary Least Squares (OLS) regression analysis are presented in Table 2. We find that the flipped classroom does not have a significant impact on the grades for the inquiry-based assignment. However, it has a positive and a significant impact on the grades for the final exam and the overall grades for the module. This may be partly attributed to the fact that the assignment was submitted in Week 6 (mid-semester), while the final exam took place in Week 12 (end of the semester). A number of studies have shown that there is a negative adjustment period associated with the "flipped" classroom format. Students have to adjust their learning tactics to suit a "flipped" classroom format [52–54]. This may also be attributed to the fact that students started putting in more work and watched the videos carefully only when the final exam drew closer.

Of all the academic control variables, I found that absences from class had a negative impact on students' performance. However, this was significant only in the case of the final exam at a 10% significant level. According to a priori expectations, studying A level economics and the time spent on independent study had a significant and positive impact on students' academic performance. In contrast, working part-time during the semester had a negative impact on students' performance in the module. However, this result was not statistically significant.

**Table 1.** Summary statistics and the difference between the flipped classroom section and the traditional lecture-based section.

| | Flipped Section (*n* = 104) | | Traditional Section (*n* = 113) | | Difference | | |
|---|---|---|---|---|---|---|---|
| | Mean | Std Error | Mean | Std Error | Mean | Std Error | t-stat |
| Standardized assignment score | 0.18 | 0.24 | −0.1 | 0.16 | 0.28 | 0.23 | 1.8 |
| Standardized exam score | 0.33 | 0.23 | −0.12 | 0.16 | 0.49 | 0.23 | 2.09 * |
| Absences | 1.46 | 0.38 | 1.05 | 0.26 | 0.41 | 0.38 | 0.79 |
| Studied A level economics | 0.16 | 0.09 | 0.14 | 0.06 | 0.02 | 0.09 | 0.21 |
| Works part-time during the semester | 0.19 | 0.09 | 0.11 | 0.06 | 0.08 | 0.09 | 0.89 |
| Average weekly time spent on independent studying (min) | 82.8 | 10.4 | 93 | 7.1 | −10.2 | 10.4 | −0.98 |

* $p < 0.1$.

**Table 2.** OLS analysis of the flipped classroom on student performance in the final exam and assignment.

| | End Semester Exam | Assignment | Aggregate Score |
|---|---|---|---|
| Flipped Classroom | 0.562 ** | 0.296 | 0.6513 *** |
| Absences | −0.106 | −0.145 * | −0.219 |
| Taken A level economics (Yes = 1) | 0.0043 ** | 0.0062 *** | 0.0075 *** |
| Part-time work (Yes = 1) | −0.1515 | −0.247 | −0.369 |
| Time spent on independent study | 0.057 *** | 0.080 | 0.042 *** |
| Constant | −8.21 * | −4.67 | −7.465 |
| Observations | 217 | 217 | 217 |
| Adjusted R squared | 0.324 | 0.31 | 0.404 |

* $p < 0.1$; ** $p < 0.05$; *** $p < 0.01$.

The results of the binary logistic regression analysis are presented in Table 3. The odds ratio is 0.621, while the inverse odds ratio (1/odds ratio) is 1.61. This shows that students in the "flipped" classroom are 1.61 times less likely to fail as compared to students in a traditional classroom. Equivalently, the odds of failing in the exam are reduced by 39% for students in the flipped format. The results reveal that demographic and household characteristics are not statistically significant. As far as the academic controls are concerned, absences from one class increase the odds of failing in the exam by 5% (odds ratio = 1.051), while an additional hour of independent studying reduces the odds of failing in the module by 4% (odds ratio = 1.041).

**Table 3.** Summary of binary logistic regression analysis for predicting the odds of failing in the module.

| Predictors | B | Std Error | Chi Square Ratio | df | *p* | Odds Ratio | Inverse Odds Ratio |
|---|---|---|---|---|---|---|---|
| Flipped = 1 | −0.476 | 0.253 | 3.53 | 1 | **0.06** | **0.621 *** | **1.61** |
| Absences | 0.05 | 0.053 | 0.858 | 1 | **0.035** | **1.051** | **0.95** |
| Taken A level economics (Yes = 1) | 0.24 | 0.289 | 0.65 | 1 | 0.041 | 1.281 | |
| Part-time work (Yes = 1) | 0.343 | 0.202 | 2.67 | 1 | 0.11 | 1.42 | |
| Time spent on independent study | −0.052 | 0.053 | 2.93 | 1 | **0.086** | **1.041 ** | **0.96** |
| Age | 0.62 | 0.295 | 4.33 | 1 | 0.037 | 1.052 | |
| Female (=1) | 0.332 | 0.119 | 2.575 | 1 | 0.101 | 1.302 | |
| White (=1) | 0.081 | 0.054 | 2.97 | 1 | 0.002 | 0.178 | |
| Mother's highest education level (Graduate and above = 1) | 0.87 | 0.153 | 3.53 | 1 | 0.001 | 1.409 | |

| Predictors | B | Std Error | Chi Square Ratio | df | *p* | Odds Ratio | Inverse Odds Ratio |
|---|---|---|---|---|---|---|---|
| Father's highest education level (Graduate and above = 1) | 0.92 | 0.153 | 3.53 | 1 | 0.001 | 1.409 | |
| Constant | −1.753 | 0.3 | 9.375 | 1 | 0.004 | 0.177 | |

\* $p < 0.1$; \*\* $p < 0.05$; R square = 0.067; −2 Log Likelihood = 546.812; $n = 250$.

## 5. Student Perceptions of the Flipped Classroom Design

The student evaluation of the flipped classroom experience was largely positive, as seen in Table 4 above. Overall, 90% of the student cohort expressed satisfaction with the "flipped" classroom model. Fifty-eight percent of the students "strongly agreed" and 40% "agreed" to a statement suggesting that the "flipped" classroom was more engaging than a traditional classroom. Eighty-two percent of the surveyed students "agreed" that the video lectures enabled them to understand the content better than the traditional classroom lecture. The positive evaluation of the flipped classroom design came forth in the open-ended question as well.

**Table 4.** Student perception of the flipped classroom, percentage of students responding.

| | | Strongly Agree | Agree | Neutral | Disagree | Strongly Disagree |
|---|---|---|---|---|---|---|
| 1 | I found the flipped classroom to be more engaging than a lecture | 58% | 40% | 14% | 5% | 0% |
| 2 | I found the video lectures helpful in understanding the content | 40% | 42% | 12% | 2% | 4% |
| 3 | I was able to watch the lectures at my own pace and time that was convenient | 33% | 47% | 13% | 6% | 0% |
| 4 | I found it useful to re-watch portions of the lecture multiple times. | 21% | 28% | 31% | 14% | 6% |
| 5 | The pre-class works takes too much time | 20% | 35% | 8% | 27% | 10% |
| 6 | I benefited from the classroom activities that allowed me to interact with my peers | 32% | 48% | 13% | 6% | 0% |
| 7 | I learnt new ways of solving the problems by observing my peers | 22% | 56% | 13% | 9% | 0% |
| 8 | I found it easier to ask questions in class | 29% | 47% | 24% | 0% | 0% |
| 9 | My interaction with the tutor improved in the flipped classroom format | 38% | 44% | 14% | 1% | 1% |
| 10 | Overall, I am satisfied with the flipped classroom format | 62% | 28% | 8% | 2% | 0% |

*"I enjoyed the structure of the module and gained a lot from the class."*

A key desirable feature of the online lectures was that they offered greater flexibility to students. Eighty percent of the students "agreed" or "strongly agreed" with the statement that "flipped" learning helped them to pace learning as per their own requirements. Fourty-nine percent of students also "agreed" or "strongly agreed" that they watched portions of the recorded lectures multiple times, which helped to enhance their understanding of the materials.

*"I could watch the videos and read at the time I felt most productive. I could follow along at my own pace and go back to the bits I found difficult. After watching the videos, the readings made more sense to me and I got a lot more from the whole process."*

One contested area in the flipped design was with respect to the time taken to do the pre-class work. Twenty percent of the students "strongly agreed" and 35% "agreed" with the statement that the "flipped" classroom took too much time.

*"It took me a lot of time, nearly 2–3 h to watch the videos and do the readings before the class each week. It left me with little time to prepare for other modules."*

This problem with the flipped classroom format has been highlighted in other studies as well [51,55]. Although, the flipped format had been designed in such a way that the workload was in consonance with a first-year module, students deemed it too high in comparison to other modules they were studying. In view of this, the weekly workload will be reviewed and updated.

One of the main advantages of the flipped classroom design is that it helps to promote peer learning. Eighty percent of the surveyed students strongly agreed or agreed that they benefitted from increased student interaction within the classroom. Additionally, 22% students "strongly agreed" and 56% "agreed" that they learnt new ways of solving problems by observing their peers. The flipped classroom format also resulted in better student–tutor interactions within the classroom. Eighty-two percent of the students stated their interaction with the tutor had improved during this process. Additionally, 76% of students also contended that they found it easier to ask questions in the "flipped" classroom model.

*"While working on the data in class, I could receive comments from the tutor at each stage. This 'hands on' approach helped to improve my understanding of the course material."*

*"I really enjoyed the pair and share activities. When the problems were hard, we helped each other until we got it right."*

*"The micro lecture helped to reinforce the key points each week. I got much more from it, since I had already watched the full lecture video in class. Also, my attention did not waver in a micro-lecture format."*

These results mirror other studies as well [27,28,51]. These studies demonstrate that students value increased peer interaction and the role of the tutor as a facilitator rather than an information deliverer in the flipped classroom format.

## 6. Discussions and Conclusions

This study demonstrated how a flipped classroom design can help to combine the benefits of the traditional lecture-based approach and a flipped class design. The micro-lectures delivered in the flipped classroom format enabled me to exercise some control over the learning process. On the other hand, the various classroom activities which were instituted in the flipped classroom format promoted greater peer interaction and also encouraged students to become active learners in this process. The online video-based lectures offered increased flexibility and motivated students to learn independently and at their own pace. The in-class problem solving enabled me to give "hands on" support to students. The research also demonstrated that the flipped classroom format resulted in the better scores in the final exam, as well as in the overall assessment for the module. In addition to this, the flipped classroom design benefitted academically weaker students and was associated with lower odds of students failing in the final exam, as compared to a traditional lecture-based format.

In terms of the theoretical framework, one can conclude that the reversal of Bloom's taxonomy helped in improving students' performance. Since the students were carrying out lower-order cognitive tasks, comprising of understanding the material at home, more time could be devoted to higher-order cognitive tasks such as application of theories to issues. Additionally, peer and group learning further

helped to improve student performance within the course. One can also see that the "flipped" classroom helped to reduce extraneous loads, as characterized in the cognitive load theory. When traditional lectures were replaced with online videos and readings, students were able to take ownership of their learning process. Learners could skip, fast forward, pause, rewind and skip any part of the online lecture, which would facilitate better management of the working memory. High-achieving students could fast forward through certain parts they clearly understood, whereas struggling students could watch the lecture multiple times. As the module tutor, I was able to analyze the course analytics for classroom quizzes to identify common areas of difficulty or clusters of expertise within the class. This enabled me to tailor activities and guidance to suit the expertise levels of students in class and facilitate better management of cognitive loads. They were thus able to concentrate better on the intrinsic load or the core concepts in economics, while reducing the extraneous loads associated with the traditional lecturing approach which do not translate into learning. On the other hand, the micro-lectures delivered in the flipped format helped to reinforce on the germane load through recapitulation of the key theories and principles.

However, the study suffers from certain limitations as well. Although, the students' performance in the final exam was better in the flipped classroom model compared to the traditional lecture-based format, there were no significant differences in the grades for the mid-term assignment. In consonance with other studies, I hypothesized that this may be attributed to the fact that students take time to adjust their learning style to a flipped format. Nevertheless, more research is needed into this subject and how to provide students with a "scaffolding" approach so that they may adjust better to a flipped classroom format. Another limitation of this study is that this study was conducted in a first-year economics classroom in a single classroom where the focus was on basic theories and principles. Questions remain on the suitability and scalability of this model in more advanced higher education settings, as well as other disciplines. The results from this study cannot be simply extrapolated to other higher education settings. This will require more empirical research in a variety of higher educational contexts and disciplines. Additionally, in this analysis I attempted to quantify the demographic and personal characteristics that influence student performance in a classroom setting. However, the students' performance may be influenced by other socio-economic, psychological and cultural contexts as well. These need to be captured through future work on the flipped classroom format.

To conclude, it can be said that, overall, the merits of this approach outweigh its potential drawbacks and the approach itself can provide an enabling classroom environment. This is especially so in areas pertaining to problem solving and development of critical thinking and quantitative skills. However, the research revealed that contestation remains with respect to students' adjustment to the flipped classroom design and the amount of pre-class work assigned within this format. A flipped classroom design cannot be implemented effectively as a "one size fits all" in all areas of higher education. It needs to be collaborated with other blended learning pedagogies to create an enabling classroom environment.

**Funding:** This research received no external funding.

**Acknowledgments:** The author is thankful to Areet Kaur and the three anonymous reviewers for their comments and feedback on the initial draft.

**Conflicts of Interest:** The author declares no conflict of interest.

**Availability of data and materials:** The data and supplementary materials supporting the manuscript are available on request.

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
