# Peer review of "“A Little Flip Goes a Long Way”—The Impact of a Flipped Classroom Design on Student Performance and Engagement in a First-Year Undergraduate Economics Classroom"

_education, doi:10.3390/educsci10110319_

Round 1

Reviewer 1 Report

This was a really interesting paper to read, and very topical. I think there's a lot of merit in publishing this paper, but I have some suggestions which I think would significantly strengthen it. I also have some concerns around the use of demographic data (protected characteristics) - these seem to have been gathered for completeness, rather than as indicated by the preceding literature review, which makes no case for demographic differences. At the same time, there is no indication of ethics approval, though the author does indicate that they attempted to conduct the study in an ethical way (I do have some reservations about coercion through awarding a 2% grade for submitting a survey though).

My other comments:

  1. Introduction. Author needs to acknowledge debate around student digital literacies; Prensky's notion of digital natives has been debunked, and so we can't make the same sweeping generalisations about 'generation Z'. Otherwise, this is a really good literature review, identifying benefits and challenges of the flipped classroom. I'd swop the paragraphs stating on lines 59 and 68 to improve the flow, but otherwise it's comprehensive and the focus of the paper is clear.
  2. Theoretical framework - again, this adds value to the paper. I like that the methodological assumptions are made clear (social constructivism and cognitive load). I'd be cautious of using the contentious term 'learning styles' and opt for 'learning preferences' instead. The diagram is helpful. I'd maybe include a reference for the (think-)pair-share technique. In the paragraph beginning line 165, the author refers to learning outcomes; these are a mix of objectives (what you want the students to do) and outcomes (what you expect students to be able to do e.g. apply) - have a think about these. I think they are more outcomes but we would word them starting with Bloom's taxonomy verbs, then have the content, then the context/delimiter. Not a big deal in the context of the paper though.
  3. Paragraph beginning line 187 - more ethical considerations would be relevant here. How were they informed e.g. plain language sheet? Was the work reviewed by an ethics panel? This is normally required for scholarship work but at the very least we need to know it was conducted in an ethical manner. 
  4. Not sure why the two groups were 120 and 130 instead of 125 and 125 but that's a minor point.
  5. You might want to clarify that the reason this is called a partial flip is the inclusion of the mini-lecture. Other than that, it looks almost wholly like a full flipped classroom. Is the mini-lecture interactive at all, to encourage active learning? If so, you could argue this was a full flipped approach.
  6. Line 212: It's PollEverywhere (https://www.polleverywhere.com).
  7. I did wonder in this section if it was worth mapping the ILOs to the assessment, to how how the assessment is relevant to them, and the subsequent learning activities?
  8. Outcomes: Here I don't fully understand the reason for the multiple regression. There is nothing in the initial lit review that suggests this is warranted. While I do believe there is merit into seeing whether previous study, part-time work, absences etc. might be relevant (and I'd expect there to be some indication that this is the case given you've investigated it), I see absolutely no rationale for bringing personal demographics into this study. I think if it had been reviewed by an ethics board this would have been picked up. Why is someone's ethnicity or parental income relevant?! The main differences I would expect from students is their readiness for independent study. It seems that your results indicate there are two groups of students - the bimodal distribution to survey item 5 suggests this is the case. Wouldn't this be a better variable to consider? And this is evidenced in the literature, to justify the comparison. The fact that there are few or no demographic differences (why would there be?) also underlines that this is irrelevant. I believe by noting gender differences you are opening up a can of worms that is not justified.
  9. In section 4, you have conflated the results and the discussion. Simply present the results here, and save the alignment with relevant literature for the discussion, which is where you need to integrate your findings with relevant literature, not just repeat the results.
  10. I'd also like to know in the methods how you analysed the open survey data; what form of thematic analysis, with an accompanying reference? Otherwise it looks as if you have cherrypicked the comments to justify your quantitative findings.

This may come across as critical, but you really need to think of the ethical issues here - asking for protected characteristics irrelevant to the inquiry, and giving them an assessment contribution for participating. Another ethical issue is that you have deliberately made one group a control group - and you need to be sure that you were not deliberately disadvantaging one group over the other.

I do otherwise think this is a very well-written study of significance and interest to the wider community. To meet the requirements for publication, you need to make some major changes to:

  1. Acknowledge contention around student digital literacies
  2. Be more transparent about the ethics
  3. Clarify what makes this a PARTIAL flipped approach
  4. Remove the demographic comparisons, as they serve no purpose to the enquiry (if the editor agrees)

Another point is that you feel the results justify your engagement with the cognitive load theory, and I'm not convinced. You may need to make the link more explicit.

Good luck with making your corrections and I look forward to seeing the revised paper, which I think will be of significant interest.

Author Response

  • The debate around digital literacies has been incorporated in the literature review (Lines 30-43).
  • Learning style has been replaced with learning preference as per the reviewer’s suggestion.
  • Pair and share activities have been explained in footnote one and a reference has been added here.

Tint, S.S. and Nyunt, E.E. (2015) Collaborative Learning with Think-Pair-Share Techniques. Computer Applications An International Journal (CAIJ) 2, 1(2015). https://journal.unnes.ac.id/nju/index.php/DP/article/view/13561.

  • “PollEverywhere,”has been put in the correct way.
  • As explained in the text, students were randomly assigned to either the flipped classroom section or the traditional classroom.Hence there were 120 students in the flipped classroom and 130 students in the traditional classroom section.
  • The process used to meet ethical guidelines of the project has been outlined, as per the reviewer’s suggestions (Lines 201-205; 267-272).
  • The coding process used for analysis of the open ended questionnaires has been clearly stipulated in Line3 258-260 and Lines 273-281.Appropriate references have been added to explain the coding process.  These comprise of:
  1. Cowan, R. L., & Fox, S. (2015). Being pushed and pulled: A model of US HR professionals’ roles in bullying situations. Personnel Review44(1), 119–139.
  2. Merriam, S. B. (2015) “Qualitative research: Designing, implementing, and publishing a study,” in V. Wang (Ed.), Handbook of research on scholarly publishing and research methods (pp.125–140). Hershey, PA:  IG Global.

  • I have taken the reviewer’s suggestion on board and rechristened the model as a “flipped” classroom model, instead of a partial flipped classroom model.

  • I have taken into account the reviewer’s suggestion and mapped the assessment against the key learning outcomes of the module and how did the in class activities facilitate the achievement of these objectives. (Lines 240-253)

  • The demographic comparisons have been removed taking account of the reviewer’s suggestions.
  • The author has organised the paper where in the results and discussion are presented in one section, the author feels that when results are discussed in the context of the larger literature at the same place, it helps to decipher the meaning more clearly. The conclusion section is not a repetition of the same discussion, rather in this section I have discussed the larger implications of the study and put it in the context of existing theoretical literature on the subject.I have also put forth the limitations of the study, as well as directions for future research in this section. 
  • I have explained the application of the flipped classroom model to the cognitive load theory in more detail (Lines 420-432).

Reviewer 2 Report

It is recommended that these references be included within the theoretical framework, because of the interest, updating and relationship with the subject matter:

Hinojo-Lucena, F.J.; Mingorance-Estrada, Á.C.; Trujillo-Torres, J.M.; Aznar-Díaz, I.; Cáceres Reche, M.P. Incidence of the Flipped Classroom in the Physical Education Students' Academic Performance in University Contexts. Sustainability 2018, 10, 1334.

Hinojo Lucena, F.J.; López Belmonte, J.; Fuentes Cabrera, A.; Trujillo Torres, J.M.; Pozo Sánchez, S. Academic Effects of the Use of Flipped Learning in Physical Education. Int. J. Environ. Public Health 2020, 17, 276.

It should be made explicit how the validation of the questionnaire is carried out, criteria for the selection of the sample, etc.

The analyses of the qualitative part should be improved with a systematic coding that shows the relationships and the proximity of relationships between them. Also the visualization of the categorical model and mind map should be improved.

In the discussion part, the interpretation of the results should be well grounded, so in general, this section is where the largest number of bibliographic citations are included. This section should be improved.

Author Response

  • The following references have been added as per the reviewer’s suggestion.
  1. Hinojo-Lucena, F.J.; Mingorance-Estrada, Á.C.; Trujillo-Torres, J.M.; Aznar-Díaz, I.; Cáceres Reche, M.P. Incidence of the Flipped Classroom in the Physical Education Students' Academic Performance in University Contexts. Sustainability 2018, 10, 1334.
  2. Hinojo Lucena, F.J.; López Belmonte, J.; Fuentes Cabrera, A.; Trujillo Torres, J.M.; Pozo Sánchez, S. Academic Effects of the Use of Flipped Learning in Physical Education. Int. J. Environ. Public Health 2020, 17, 276.
  • The process used for administering the questionnaires and recruiting students for participating in this research study has been clearly explained (Lines 263-271).

  • The coding process used for analysis of the open ended questionnaires has been clearly stipulated in lines 287-295.

  • The author feels that the presentation of the two diagrams is clear, there have also been explained in the subsequent sections, and a justification has been provided for these.

  • The discussion part has been revised as per the author’s suggestions.

Reviewer 3 Report

The paper tackles the issue of the flipped classroom, as an active learning pedagogy to engage Z generation of students. The topic itself seems interesting.

However, there are some concerns in my review that the author should modify as follows:

-     The abstract of this paper needs further improvement, enable readers to understand the highlights through the abstract. 

-     Introduction and literature are developed well but as I see there are not any RQ and addressed motivation. Motivation should be well-grounded in the introduction section.  Moreover, the literature review section should be updated with the 2018, 2019 and 2020 papers.

-  The author should reformulate ‘in this study, I experimented’, ….’I’, …’I’ in an impersonal way.

Author Response

  1. The abstract has been modified as per the reviewer’s suggestion (Lines 7-25).
  2. “This study I experimented with…”, this line has been re-formulated to “This study experiments with…” (Lines 10-11 in the revised script).
  3. The main research questions and motivation of the study have been clearly highlighted in the introduction section (Lines 118-128).
  4. The literature review has been updated by adding the following references:
  • Hinojo-Lucena, F.J.; Mingorance-Estrada, Á.C.; Trujillo-Torres, J.M.; Aznar-Díaz, I.; Cáceres Reche, M.P. Incidence of the Flipped Classroom in the Physical Education Students' Academic Performance in University Contexts. Sustainability 2018, 10, 1334.
  • Hinojo Lucena, F.J.; López Belmonte, J.; Fuentes Cabrera, A.; Trujillo Torres, J.M.; Pozo Sánchez, S. Academic Effects of the Use of Flipped Learning in Physical Education. Int. J. Environ. Public Health 2020, 17, 276.
  • Gómez-García, G.; Marín-Marín, J.A.; Romero-Rodríguez, J.-M.; Ramos Navas-Parejo, M.; Rodríguez Jiménez, C. Effect of the Flipped Classroom and Gamification Methods in the Development of a Didactic Unit on Healthy Habits and Diet in Primary Education. Nutrients202012, 2210.

Round 2

Reviewer 1 Report

Well done to the author for their quick turnaround of their paper in response to reviewer comments. I am satisfied that my comments about flipped (versus partially flipped), student digital literacies, ethics, and demographic comparisons have all been appropriately addressed. It is also good to see additional attention paid to qualitative thematic analysis of open survey responses. Well done again.